# Spike-Dependent Dynamic Partitioning of the Locus Coeruleus Network through Noradrenergic Volume Release in a Simulation of the Nucleus Core

**DOI:** 10.3390/brainsci12060728

**Published:** 2022-06-01

**Authors:** Shristi Baral, Hassan Hosseini, Kaushik More, Thomaz M. C. Fabrin, Jochen Braun, Matthias Prigge

**Affiliations:** 1Research Group Neuromodulatory Networks, Leibniz Institute for Neurobiology, 39118 Magdeburg, Germany; shristi.baral@st.ovgu.de (S.B.); seyyedhassan.hosseini@lin-magdeburg.de (H.H.); kaushik.more@med.ovgu.de (K.M.); fabrintmc@gmail.com (T.M.C.F.); 2Cognitive Biology, Faculty of Natural Sciences, Otto-von Guericke University, 39118 Magdeburg, Germany; 3Center for Behavioral Brain Sciences, 39118 Magdeburg, Germany

**Keywords:** Locus coeruleus, volume diffusion, noradrenaline, non-synaptic connectivity, alpha2-adrenergic receptors

## Abstract

The Locus coeruleus (LC) modulates various neuronal circuits throughout the brain. Its unique architectural organization encompasses a net of axonal innervation that spans the entire brain, while its somatic core is highly compact. Recent research revealed an unexpected cellular input specificity within the nucleus that can give rise to various network states that either broadcast norepinephrine signals throughout the brain or pointedly modulate specific brain areas. Such adaptive input–output functions likely surpass our existing network models that build upon a given synaptic wiring configuration between neurons. As the distances between noradrenergic neurons in the core of the LC are unusually small, neighboring neurons could theoretically impact each other via volume transmission of NE. We therefore set out to investigate if such interaction could be mediated through noradrenergic alpha2-receptors in a spiking neuron model of the LC. We validated our model of LC neurons through comparison with experimental patch-clamp data and identified key variables that impact alpha2-mediated inhibition of neighboring LC neurons. Our simulation confirmed a reliable autoinhibition of LC neurons after episodes of high neuronal activity that continue even after neuronal activity subsided. Additionally, dendro-somatic synapses inhibited spontaneous spiking in the somatic compartment of connected neurons in our model. We determined the exact position of hundreds of LC neurons in the mouse brain stem via a tissue clearing approach and, based on this, further determined that 25 percent of noradrenergic neurons have a neighboring LC neuron within less than a 25-micrometer radius. By modeling NE diffusion, we estimated that more than 15 percent of the alpha2-adrenergic receptors fraction can bind NE within such a diffusion radius. Our spiking neuron model of LC neurons predicts that repeated or long-lasting episodes of high neuronal activity induce partitioning of the gross LC network and reduce the spike rate in neighboring neurons at distances smaller than 25 μm. As these volume-mediating neighboring effects are challenging to test with the current methodology, our findings can guide future experimental approaches to test this phenomenon and its physiological consequences.

## 1. Introduction

The Locus coeruleus (LC) is a brain stem nucleus that is the primary source of norepinephrine (NE) in the central nervous system. It comprises a unique neuronal architecture with an extensive and diffuse axonal innervation throughout the entire brain and spinal cord, while the somatic core consisting of several hundred neurons is highly compact [1]. Research on the noradrenergic system has revealed astonishing insights on how a single neuromodulator such as NE can pointedly impact diverse brain functions ranging from arousal to the sensation of pain to memory encoding and sensory processing [2,3,4,5]. In addition, much research is focused on understanding how expression profiles of adrenergic receptors in their respective axonal terminal fields are correlated to mental disorders such as attention deficit, clinical depression, or drug abuse—reviewed in [6]. Distinct catecholaminergic receptor subtypes can be activated through a dynamic modulation of an extracellular NE gradient in the vicinity of a synaptic release site [4,7]. In this respect, the type of NE release site in axonal terminals has been controversially discussed in the past; to one extreme, a pure volume transmission was assumed as main mode of action, and, on the other extreme, a purely wiring-like transmission [8,9,10,11,12].

On the opposite, less research has focused on the synaptic wiring modes within the LC core itself. Primarily, such research interest was hampered by the canonical view that LC activity is mainly based on autonomous pacemaking of a homogenous noradrenergic cell population. Yet, recent technological advancements in single-cell genetics, viral tracing, and optical manipulation of neurons have begun to challenge this view. Already a surprising heterogeneity in the LC cell population regarding their projection-specificity and physiological function has been described [3,13,14]. Nevertheless, we are still missing a framework on how individual LC neurons are computing their brain-wide inputs to the nucleus and how this information is relayed within the nucleus and thereby to the entire brain [6]. As the somas in the nucleus are in unusually close proximity to each other, volume-mediated interactions are perceivable. Previous work on catecholaminergic neurons in the substantia nigra and ventral tegmental area argue for such volume interaction between neurons as a key mechanism to fully describe neuronal activities in these areas [15,16,17]. Early work in LC focused primarily on electronic coupling via the gap junction as the main mode of connection [18]. Gap junctions are most prominent in the dendritic compartment and decrease in abundance with aging [19,20]. They give rise to subthreshold oscillation that can synchronize the entire nucleus during postnatal periods [18]. Yet, recent high-density silicon probes recordings confirm that at older age neurons from different subsets of the LC exhibit asynchronous activity [21]. Here, correlated activity patterns were observed within these subsets of LC neurons at different timescales: from sub-milliseconds to milliseconds to periodic oscillation in the range of seconds. Such a wide range of temporal interactions between LC neurons is beyond a pure gap junction-coupled neuronal network.

Indeed, several neuromodulators such as galanin, serotonin, neuropeptides, and NE itself are known to also impact LC network activity on the timescale of seconds and even minutes [22,23,24,25]. Electrophysiological studies clearly demonstrated a strong NE-mediated control of the LC network via high-affinity alpha2-receptor [26,27,28,29]. Infusion of antagonists leads to hyperpolarization that is mediated through a NE-sensitive potassium conductance [30,31,32]. This decrease in excitability is mediated via G_i/o_ pathways that trigger the G-protein coupled inward rectifying potassium channel GIRK2 (also called Kirk3) and thereby controlling afferent activation of LC neurons as well as the network state of the entire nucleus [25,33,34,35,36]. So far, these questions have been solely approached through a drug-based experimental design, where a drug effect is either studied with conventional patch-clamp recordings on a single cell level or, at the other extreme, on the level of the entire LC network. As our view of the LC network shifts to a microcircuitry level of functional specific subnetworks within the LC, spiking patterns of individual neurons in their respective subnetworks become more critical for our understanding. Episodes of high spiking rate (burst-like activity) in individual neurons can evoke elevated intracellular calcium levels in soma that surpass the baseline level seen during tonic activity [37]. These high calcium levels can trigger the release of NE-containing vesicles from dendritic and somatic compartments [37,38]. It is challenging to evaluate how synaptic and extracellular NE levels influence the LC network and neighboring neurons with the current technologies. We, therefore, set out to employ a Hodgkin–Huxley-based model to study NE-mediated local effects with the nucleus.

## 2. Methods

### 2.1. Acute Brain Slice Preparation

All procedures were approved by the Landesverwaltungsamt/Halle and conformed to German regulation. All studies were performed in compliance to animal protocol 42502-2-1545 LIN. Acute brain slices for electrophysiology were obtained from Ai9 × DbH-Cre mice (Jax# 007905 and 033951) on a C57Bl6/J background. For tissue clearing, mice from an Ai9 × PV-Cre were used (Jax# 007905 and 17320).

Mice were injected intraperitoneally with pentobarbital (Narcoren®, Boehringer Ingelheim Vetmedica GmbH, D-55216 Ingelheim, Germany,130 mg/kg, i.p.) and perfused with a carbonated (95% O_2_, 5% CO_2_) ice-cold NMDG-HEPES aCSF (in mM: 92 NMDG, 2.5 KCl, 1.25 NaH_2_PO_4_, 30 NaHCO_3_, 20 HEPES, 25 glucose, 2 thiourea, 5 Na-ascorbate, 3 Na-pyruvate, 0.5 CaCl_2_·2 H_2_O, and 10 MgSO_4_·7H2O; pH adjusted to 7.3–7.4 and osmolality measured and adjusted to 300–310 mOsm/l). After decapitation, 300-μm coronal brain stem slices were prepared in carbonated ice-cold NMDG-HEPES aCSF using a vibratome (Leica VT 1200S) and allowed to recover for 10 min at 33 °C in NMDG-HEPES aCSF. Afterwards slice were places in HEPES holding aCSF (in mM: 92 NaCl, 2.5 KCl, 1.25 NaH_2_PO_4_, 30 NaHCO_3_, 20 HEPES, 25 glucose, 2 thiourea, 5 Na-ascorbate, 3 Na-pyruvate, 2 CaCl_2_·2 H_2_O, and 2 MgSO_4_·7 H2O. pH titrated to 7.3–7.4 with several drops of concentrated 10 N NaOH, and osmolality measured and adjusted to 300–310 mOsmo/l). Subsequently, slices were kept at RT in carbonated HEPES holding aCSF (in mM: 92 NaCl, 2.5 KCl, 1.25 NaH2PO4, 30 NaHCO3, 20 HEPES, 25 glucose, 2 thiourea, 5 Na-ascorbate, 3 Na-pyruvate, 2 CaCl_2_·2 H_2_O, and 2 MgSO_4_·7 H_2_O. pH titrated to 7.3–7.4 with several drops of concentrated 10 N NaOH, and osmolality measured and adjusted to 300–310 mOsmol/kg) and trasnfered transferred to recording aCSF (in mM): 124 NaCl, 2.5 KCl, 1.25 NaH_2_PO_4_, 24 NaHCO_3_, 12.5 glucose, 5 HEPES, 2 CaCl_2_·2H_2_O, and 2 MgSO_4_·7 H_2_O. Titrated pH to 7.3–7.4 with a few drops of concentrated 10 N NaOH, osmolality measured and adjusted to 300–310 mOsm/l) for use. The recording chamber was perfused with carbonated recording aCSF at a rate of 2 mL min^−1^ and maintained at 32 °C.

### 2.2. Brain Slice Electrophysiology

Whole-cell patch-clamp recordings were performed under visual control using oblique illumination on a Slicecope Pro2000 (Scientifica, Uckfield, East Sussex, UK) equipped with a 12-bit monochrome CMOS camera (Hamamatsu Model OrcaFlash). Borosilicate glass pipettes (Sutter Instrument BF100-58-10) with resistances ranging from 3–7 MΩ were pulled using a laser micropipette puller (Sutter Instrument Model P-2000). Pipettes were filled using a standard intracellular solution (in mM: 135 potassium gluconate, 4 KCl, 2 NaCl, 10 HEPES, 4 EGTA, 4 Mg-ATP, 0.3 Na-GTP; 280 mOsm kg^−1^; pH adjusted to 7.3 with KOH). Whole-cell voltage-clamp recordings were performed using a MultiClamp 700B amplifier, filtered at 8 kHz and digitized at 20 kHz using a Digidata 1550A digitizer (Molecular Devices).

### 2.3. HEK Cells Electrophysiology

Human embryonic kidney (HEK) cells stably expressing G-protein rectifying potassium channels (GIRK 1/2 subunits), kindly provided by Dr. A. Tinker UCL London, GB, and Dr. S. Herlitze, were maintained at 37 °C in Dulbecco’s modified Eagle’s medium, 4.5 g/L D-glucose, supplemented with 10% fetal bovine serum (Gibco) and penicillin/streptomycin incubated under 5% CO_2_. Cells were cultured in a 24-well plate, on 12 mm glass coverslips.

Whole-cell patch-clamp recordings were performed under visual control using a Slicecope Pro2000 (Scientifica, Uckfield, East Sussex, UK) equipped with a 12-bit monochrome CMOS camera (Hamamatsu Model OrcaFlash). Borosilicate glass pipettes (Sutter Instrument BF100-58-10, 2–5 MΩ) were pulled using a laser micropipette puller (Sutter Instrument Model P-2000). Pipettes were filled using the internal solution as follows: 100 mM potassium aspartate, 40 mM KCl, 5 mM MgATP, 10 mM HEPES-KOH, 5 mM NaCl, 2 mM EGTA, 2 mM MgCl_2_, 0.01 mM GTP, pH 7.3 (KOH). The bath was filled using the following external solution: 20 mM NaCl, 120 mM KCl, 2 mM CaCl_2_,1 mM MgCl_2_, 10 mM HEPES-KOH, pH 7.3 (KOH). Whole-cell voltage-clamp recordings were performed using a MultiClamp 700B amplifier, filtered at 8 kHz and digitized at 20 kHz using a Digidata 1550A digitizer (Molecular Devices). The following protocol was used during the IV curve recordings: fifteen sweeps starting from −80 mV (Δ +10 mV). The cells were clamped on a holding potential at −60 mV.

### 2.4. Immunohistochemistry

Mice were transcardially first perfused with phosphate-buffered saline (PBS 1×) and then with 4% paraformaldehyde. The brains were extracted and stored overnight in 4% PFA followed by soaking in 30% sucrose to avoid forming water-crystals during slice sectioning at 4°C. Brains were sliced on a microtome (LEICA SM2010R) to 40 μm thick coronal sections. Afterward, brain slices were first permeabilized and blocked by incubating in PBST (Phosphate-buffered saline with 0.3% Triton-X) containing 10% horse serum for 2 h at room temperature to prevent non-specific binding of antibodies. Sections were incubated in a solution containing the primary antibodies, 2% horse serum, and 0.3% Triton-X for 24 h at 4 °C. The primary antibodies used were guinea pig anti-TH (Synaptic Systems, Catalog No. 213104) with 1:500 dilution and rabbit anti-DBH (ImmunoStar, Catalog No. 22806) with 1:1000 dilution. Following three washes in 1× PBS for 10 min, the sections were then incubated in a solution containing Cy3 anti-guinea pig IgG, (Jackson Immuno Research, Catalog No. 706-165-148, West Grove, PA, USA), Cy5 AffiniPure Donkey Anti-Rabbit IgG, (Jackson Immuno Research, Catalog No. 711-175-152) both in 1:200, 5% horse serum and 0.05% Triton-X for 2 h at room temperature. The sections were then washed thrice with 1× PBS for 10 min and then incubated in a solution containing DAPI at a dilution of 1:30,000. All the above-mentioned solutions were made in 1× PBS. The sections were mounted on glass slides and coverslips containing DPX mounting media. Images were acquired on a Leica SP8 confocal microscope using 63× oil-immersion objectives (NA 1.4). Images were analyzed using ImageJ (https://imagej.nih.gov/ij/, accessed on 12 May 2022). For images in Figure 1, we overlaid the reference atlas form The Mouse Brain in Stereotaxic Coordinates 3rd Edition Franklin & Paxinos at the anterior–posterior axis, −5.40 mm.

### 2.5. Tissue-Clearing

Sectioning of 4% PFA fixated mouse brains were performed in a coronal configuration using a brain mold to obtain 2-mm thick slices that contain the major part of the LC nucleus (AP 4.4 to 6.4 mm) (brain matrix, 3D printed). Brain slices were then washed in PBS for 1 h and incubated at different series of mixtures prepared from tetrahydrofuran (THF, Sigma-Aldrich, Catalog No.186562-1L, St. Louis, MO, USA): dH_2_O + 2% Tween-20 to dehydrate brain slices. The pH of the THF solution was adjusted to 9.0 by adding triethylamine. The slices were washed in 50% THF solution for 4 h at 4 °C followed by 70%, 90%, and 100% THF solutions under the same conditions. To ensure complete dehydration, we washed the section with 100% THF twice. Slices were then bleached in chilled 5% H_2_O_2_ in THF overnight at 4 °C. The next day, the slices were rehydrated in the reverse order by washing them in 100%, 90%, 70%, and 50% THF solutions as above for 4 h at 4 °C. Following this, slices were washed twice in a PBS/0.2% TritonX for 1 h each wash. Slices were then permeabilized in PBS/0.2% TritonX/20% DMSO/0.3 M glycine at 4 °C for 2 days and then blocked in PBS/0.2% TritonX/10% DMSO/6% horse serum at 4 °C for another 2 days. Samples were kept in a primary antibody (guinea pig anti-TH Synaptic Systems, Catalog No. 213104, Göttingen, Germany) dilution in PBS/0.2% Tween-20/5% DMSO/3% horse serum with 10 µg/mL heparin for two days at 4 °C. Primary antibodies were diluted at 1:400. Following this treatment, samples were washed in PBS/0.2% Tween-20 with 10 µg/mL heparin for 1 h at 4 °C for 4× and then kept in a secondary antibody (Cy3 anti-guinea pig IgG, Jackson Immuno Research, Catalog No. 706-165-148) in 1:150 dilution in PBS/0.2% Tween-20/3% horse serum with 10 µg/mL heparin at 4 °C for 2 days and washed in PBS/0.2% Tween-20 with 10 µg/mL heparin for 1 h at 4 °C for 4×. For imaging, samples were embedded in 0.8% phytagel and dehydrated in the THF/dH20 series exactly as described before. After the dehydration, the samples were transferred to ethyl cinnamate (ECi) at room temperature for clearing. ECi was replaced with fresh ECi the following day, and samples were stored at room temperature.

### 2.6. Light-Sheet Microscopy

The light-sheet microscopy was performed using LaVision Biotech Ultramicroscope II (Miltenyi Biotec, Bergisch-Gladbach, Germany), which was equipped with a Zyla 4.2 PLUS sCMOS camera (Oxford Instruments, Abingdon-on-Thames, UK). The embedded brain slice was placed in the sample chamber containing ethyl cinnamate (ECi). The brain volume was sampled through a 4× objective (LaVision BioTec LVMI-Flour 4×/0.3). The objective was configured to match the refractive index of ECi (1.558), for optimal imaging. For excitation, we used light from a EXW-12 extreme laser (NKT Photonics, Birkerød, Denmark), which was guided through the appropriate excitation filters (AHF Analysentechnik AG, Tübingen, Germany) for Cy3 fluorescence. We started acquiring the image using ImSpector software (Miltenyi Biotec, Bergisch-Gladbach, Germany) after setting the parameters such as sheet width, z-step sizes, number of tiles, and zoom. For post-processing and segmentation, we used Imaris software (version 9.72 Bitplane, Zurich, Switzerland).

### 2.7. Modeling

Based on cortical membrane kinetics, Alvarez [39] proposed a two-compartment mathematical model of LC cells, with soma and dendrite connected via a gap junction on the dendrite. We employed this model as an initial starting point and extended the model to incorporate quantal release of NE, which eventually leads to the opening of GIRK channels and inhibition of the neighboring cell and/or itself via alpha2-receptor located in the somatic area. The release event is coupled with the amount of cytosolic calcium [Ca^2+^] in the cell.

### 2.8. Dynamic Equation of State Variables

Voltage change in the soma is driven by the balance of currents, including the electrode current *I_elec_*, the HH-sodium current *I_Na_*, the HH-potassium current *I_K_*, the persistent sodium current *I_P_*, the Ca^2+^-driven potassium current *I_AHP_*, the voltage-driven calcium current *I_Ca_*, the synaptic current from GIRK conductance *I_GIRK_*, the current from dendrite *I_dend_*, the leak current *I_L_*, and the independent random current *η*(*t*).
(1)$$\begin{array}{c} \;\;\frac{dV_{S}^{i}}{dt}c_{m}=I_{elec}-I_{Na}-I_{K}-I_{P}-I_{AHP}-I_{Ca}-I_{GIRK}-I_{dend}-I_{L}+\eta \left(t\right)\\ \frac{\mu F}{area}\frac{mV}{ms}=\frac{\mu A}{area}=\frac{mS}{area}\;\mathrm{mV},\;\;area\;\mathrm{in}\;\left({\mathrm{cm}}^{2}\right) \end{array}$$

Voltage changes in the dendrites are driven by current from soma *I_soma_*, gap junctions from other dendrites *I_GJ_* and dendritic leak current *I_LD_*
(2)$$c_{m}\frac{\;dV_{D}^{i}}{dt}\;=-I_{soma}-I_{GJ}-\;I_{LD}$$
(3)$$I_{AHP}=g_{AHP}\;r\;\left(V-V_{K}\right)$$

The Ca^2+^ concentration changes over time as a function of voltage.
(4)$$r\;=\;\frac{\left[Ca\right]}{\left[Ca\right]\;+\;1\;\mu M}$$

Specifically, membrane depolarization during a spike allows Ca^2+^ to enter the neuron. As described above, the opening and closing of the *g_AHP_* potassium conductance shadows the cytosolic Ca^2+^ concentration.

The steady-state of the calcium is
(5)$${\left[Ca\right]}_{\infty }=f_{in}\left(V\right)\tau_{Ca}\;=\;\frac{f_{ca}\;k_{Ca}\left(V-V_{Ca}\right)}{1+\exp\left(-\frac{\left(V\;+\;25\right)}{2.5}\right)}\;\;,\;\;k_{Ca}\;=2,\;f_{Ca}=\;0.002\frac{\mu M}{\mathrm{mVms}}$$

And the update rule is
(6)$${\left[Ca\right]}_{\infty }=\frac{M_{Ca}}{132.6\;\mathrm{mV}}\;\frac{V_{Ca}-V}{1+\exp\left(-\frac{\left(V\;+\;25\right)}{2.5}\right)}$$
(7)$${\left[Ca\right]}_{i\;+\;1}={\left[Ca\right]}_{\infty }+\left({\left[Ca\right]}_{i}-{\left[Ca\right]}_{\infty }\right)\exp\left(-\frac{dt}{\tau Ca}\right)$$

As the voltage-dependent factor in the expression of ${\left[Ca\right]}_{\infty }$ peaks at 132.6 mV for *V_Ca_* = 120 mV, it is convenient to define a new parameter, *M_ca_* = 42.4 μM and $\tau_{Ca}$ = 80 ms (Appendix A).

### 2.9. Dendro-Somatic and Soma-Somatic NE Release and Alpha2-Receptor Activation, and GIRK Conductance

Action potentials in the presynaptic neuron results in NE release of dense core vesicles from presynaptic sites, which are found at the soma and at nearby dendrites. The release fraction *R_rel_* depends on the level of presynaptic calcium at the time of the presynaptic action potential.
(8)$$R_{rel}\left(\left[Ca_{AP}^{j}\right]\right)=\;\frac{2\cdot {\left[Ca_{AP}^{j}\right]}^{k}}{{\left[Ca_{high}\right]}^{k}+\;{\left[Ca_{AP}^{j}\right]}^{k}},\;k=1,\;\left[Ca_{high}\right]=1.3$$

$\left[Ca_{AP}^{j}\right]$ is presynaptic cytosolic calcium at the time of the action potential. [*Ca_high_*] = 1.3 μM is the amount of cytosolic calcium accumulated when the cell fires at 20 Hz. The exponent, *k* = 1, modulates the release.

Once released, NE binds to alpha-2 receptors in the postsynaptic membrane, which through G-protein signaling opens an inwardly rectifying potassium channel (GIRK: G-protein-coupled inward-rectifying potassium channel). The effective time course of the opening and closing of this synaptic conductance is comparatively slow [40]. We approximated this time-course with a double-exponential function with *τ_fast_* = 300 ms and *τ_slow_* = 350 ms as well as measured the voltage-dependence of the GIRK current in a HEK cells line that was stably expressing GIRK1/2 subunit channel (Appendix A) [40,41]
(9)$$\begin{array}{c} P_{open}\left(t\right)=B\left[f_{slow}^{j}\left(t\right)-f_{fast}^{j}\left(t\right)\right]\\ f_{fast}^{j}\left(t\right)=\exp\left(\frac{-t-t_{AP}^{j}}{\tau_{fast}}\right)\\ f_{slow}^{j}\left(t\right)=\exp\left(\frac{-t-t_{AP}^{j}}{\tau_{slow}}\right)\\ t_{peak}=\frac{\tau_{fast}\tau_{slow}}{\tau_{slow}-\tau_{fast}}\log\frac{\tau_{slow}}{\tau_{fast}}\\ B=\frac{1}{e^{\frac{-t_{peak}}{\tau_{slow}}}-e^{\frac{-t_{peak}}{\tau_{fast}}}}\\ B=\left[{\left(\frac{\tau_{fast}}{\tau_{slow}}\right)}^{\left(\frac{\tau_{fast}}{\tau_{slow}-\tau_{fast}}\right)}-{\left(\frac{\tau_{fast}}{\tau_{slow}}\right)}^{\left(\frac{\tau_{slow}}{\tau_{slow}-\tau_{fast}}\right)}\right] \end{array}$$

The absolute value *G_rect_* of this rectifying conductance was not constant but changed with the postsynaptic membrane potential. Accordingly, we needed to consider this factor in the equation.
(10)$$G_{rect}\left(V^{i}\right)=\frac{0.2}{1+\exp\left(0.05\left(V^{i}-V_{GIRK}\right)\right)}$$

*G_rect_* is given in μS per mm^2^.

Combining all three factors for multiple successive presynaptic action potentials *AP*_1_, *AP*_2_, …, the GIRK conductance is given then by:(11)$$g_{GIRK}^{ij}\left(V^{i},t,\left[Ca_{AP1}^{j}\right],\left[Ca_{AP2}^{j}\right]\ldots \right)=G_{rect}\left(V^{i}\right)\left[P_{open}\left(t-t_{AP1}\right)R_{rel}\left(\left[Ca_{AP1}^{j}\right]\right)\right]+\left[P_{open}\left(t-t_{AP2}\right)R_{rel}\left(\left[Ca_{AP2}^{j}\right]\right)+\ldots \right]$$

It was released from or near soma and bound near soma. This hyperpolarized the membrane potential of the neighboring soma. *S_ij_* is the strength of the synaptic connection from neuron *j* to neuron *i*.
(12)$$\begin{array}{ll} I_{GIRK}^{i} & =\left(V_{s}^{i}-V_{GIRK}\right)\cdot g_{GIRK}^{ij}\cdot \underset{j}{\sum }S_{ij}\left(V_{s}^{i},t,\left[Ca_{AP1}^{j}\right],\left[Ca_{AP2}^{j}\right]\ldots \right)\\ & =\left(V_{s}^{i}-V_{GIRK}\right)G_{rect}\left(V_{i}\right)\left(\underset{j}{\sum }\left[S_{ij}\left[P_{open}\left(t-t_{AP1}\right)R_{rel}\left(\left[Ca_{AP1}^{j}\right]\right)+P_{open}\left(t-t_{AP2}\right)R_{rel}\left(\left[Ca_{AP2}^{j}\right]\right)+\ldots \right]\right]\right) \end{array}$$

### 2.10. NE Diffusion

To estimate the effect of NE released into the extracellular space, we computed the time-course of the NE concentration at some distance from a release site as by:(13)$$C_{NE}\left(r,t\right)=\frac{\mathrm{Exp}\left(-\frac{r^{2}}{4\;D\;t}-k\;t\right)}{\alpha {\left(4\;D\;\pi \;t\right)}^{3/2}}$$
where *r* is distance; *t* is time; *D* = 3.4 × 10^−6^ cm^2^/s is the diffusion constant; and *k* = 20 Hz is the reuptake constant [42]. The fraction of NE-binding alpha2-receptors was computed from
(14)$$\tau_{a2R}\;\frac{dx}{dt}=-x+x_{a2R},\;\tau_{a2R}=\frac{1}{200\;\mathrm{nM}\;k_{on}+k_{off}},\;x_{a2R}=\frac{200\;\mathrm{nM}\;k_{on}}{200\;\mathrm{nM}\;k_{on}+k_{off}},$$
where *k_on_* = 1.0 and *k_off_* = 1.25 nM are the kinetic constants for NE binding and unbinding, respectively.

The resulting time course and voltage dependence of GIRK conductance was fitted with a double-exponential function with *τ_fast_* = 1400 ms and *τ_slow_* = 1450 ms and an attenuation factor of 0.07 (Appendix A).

To take into account the kinetics of transmitter binding and release, the resulting time-course was convolved with a low-pass filter (“alpha function”) $g\left(t\right)=\frac{t}{\tau }\exp\left(-t/\tau \right)$ with characteristic time τ=50 ms.

### 2.11. Data Analysis

All data were analyzed in Python. For extraction of action potential features (simulated and experimental), we used the Allen Institute ephys package. The generalized leaky integrate and fire models (GLIF) were applied. Additional analyses were performed using custom Python scripts, all of which are available on the laboratory GitHub account.

## 3. Results

We first set out to confirm that LC neurons express both key enzymes to catalyze NE in somatic and dendritic compartments throughout their lifetime. Immunostaining against tyrosine hydroxylase (TH), as well as dopamine-beta-hydroxylase (DbH) in young mice (P14) and old mice (older than P500), revealed expression of both enzymes in all cells at both ages (Figure 1a,b). Both stainings also highlight the compact nature of the LC nucleus itself. Furthermore, a dense dendritic arbor directed towards the medial-rostral axis becomes visible [18]. Image overlay of the fluorescence channel from TH and DbH confirms an overall co-expression of both enzymes throughout the somatic compartment within the LC nucleus (see Figure 1a,b, overlay). Surprisingly, we observed differential spatial expression levels of TH and DbH in the dendritic arbor. Here, both poles of the LC (dorsal and ventral) appeared to have protruding dendrites that mainly display a TH immunosignal. The central part of the nucleus seems to extend dendrites that preferentially express DbH (Figure 1c, 3 animals). The dendritic expression profile for the DbH enzymes was also similar for young and old mice along the anterior–posterior axis (Appendix A)

As it is difficult to experimentally approach somatic or dendritic NE release from a single LC neuron and investigate how it affects spike activity in neighboring LC neurons, we decided to develop a computational model that allows us to study three modes of NE-mediated interactions. The first mode was via dendro-somatic synapses [12]; secondly, we considered a somatic autoinhibition [29,43] and, thirdly, we investigated if a NE volume diffusion from a somatic release point can lead to an inhibition of a neighboring neuron.

To realistically model the distribution of the distances between LC neurons, we employed a tissue clearing approach to map the positions of noradrenergic neurons in the LC. First, we prepared 2-mm thick brain stem slices from PV-Cre × Ai9 animals. We then used a modified iDISCO protocol to clear the entire brain stem slices and image the slice volume via fluorescence light sheet microscopy (see Methods for details). After acquiring images of the entire brain stem, we visualized anti-TH+ and PV-dTomato neurons with high contrast (Figure 2a). We segmented the LC and resolved positions of 220 to 439 anti-TH+ neurons (4 nuclei, 2 animals) (Figure 2b). Based on the x-y-z coordinates of individual anti-TH+ neurons, we computed the nearest-neighbor (NN) distances for each cell in all four analyzed nuclei. The distribution of the NN-distances peaked at distances of beta-hydroxylase around 25 μm (median 41.34 μm) (see Figure 2c).

To investigate the possible effects of a noradrenergic interaction between LC neurons on a single cell level, we chose as a starting point a previously published mathematical model that describes the electronic coupling between two LC neurons via gap junctions [39] This model recapitulates frequency-dependent synchronization between LC neurons as an electrotonic coupling strength between dendrites. Here, strong gap-junction coupling between dendritic compartments, as described experimentally in young mice [18,19,20], led to synchronization of spontaneous spiking for all frequencies, while weak dendritic coupling promotes spike synchrony only at higher spiking frequencies. Comparison of our initial model under a weak electrotonic coupling regime reiterates our current-clamp recordings from noradrenergic neurons from acute brain slices prepared from mice older than P70. The spontaneous spike rate at the resting potential was 4.2 ± 1.77 Hz (*n* = 6 neurons, 2 animals), while our model estimated 4.3 ± 0.03 Hz (Figure 3a). The overall features of action potentials between our model and experimental data largely coincided (Figure 3b). To emulate different excitability levels of LC neurons, we recorded the spike rate during a three hundred millisecond long current pulse at various amplitudes (Figure 3c, upper trace). Additionally, in our model, the current steps evoked an increased spike rate (Figure 3c, lower trace). As expected, spike rates increased with amplitudes of injected current. Quantification of the spike rate at various current step amplitudes revealed an overall linear dependence in the frequency–current relation (FI-curve) for experimental and simulated data (compare Figure 3d1,d2). Interestingly, we observed a higher spike rate for the two initial spikes and an overall steeper slope in the FI-curve for these early spikes compared to the following spikes (Appendix A). We also found such an increased spike rate for the initial spikes in our model; this effect gets more pronounced at higher current amplitudes for experimental and simulated data (compare Appendix A).

To simulate a noradrenergic modulation between LC neurons, we extended our model with a calcium-dependent release of NE. We considered a NE release site as a point source of NE secretion that either mimics a dendro-somatic synapse or a somatic release site that can either retroact on the secreting neuron (self-inhibition) or effect neighboring neurons via diffusion (neighboring inhibition). High spiking rates lead to membrane fusion of NE-containing large-dense vesicles at the soma [37,38]. The voltage-dependent rise in calcium is modeled by Equations (3)–(7) (see Methods). Based on amperometric recordings in acute brain slices, we assumed an initial extracellular NE concentration after vesicle fusion of 200 nM [37]. We combined the time courses of several molecular events in Equation (11), i.e., binding of NE to alpha2-receptors, initiation of G_i/o_ signaling cascade, and the subsequent release of G_βγ_ dimer that increases GIRK channel conductance. The dependence of GIRK conductance and intracellular calcium levels is shown in Figure 4a. For a given calcium level (dotted line in Figure 4a), the temporal development of GIRK open probability (*P_open_*) is modeled based on electrophysiological characterization of GIRK current in LC neurons (Appendix A) [40].

Based on these assumptions, we first tested our model for a synaptic-like dendro-somatic interaction between two LC neurons. To mimic episodes of high activity, we injected current amplitudes for 1-sec that led to spike rates close to 20 Hz in neuron 1 (N1) (Figure 4c1). As expected, an increased spike rate led to an uprise of intracellular calcium levels (Figure 4c1, magenta trace and Appendix A). As described in Equation (5), this increase translated into a delayed, slowly developing onset of GIRK conductance Equation (11) (Figure 4b,c1, blue trace). The generated GIRK conductance decreased with Tau = 350 ms after the current injection step ended. In our model, the increased GIRK conductance during and after episodes of high spike activity led to complete inhibition of spontaneous spiking in the dendro-somatically connected neuron 2 (N2). Only after GIRK conductance returned to basal levels, N2 resumed spontaneous spiking.

In a different scenario, we investigated how the somatic release of NE impacts the neuronal activity of the releasing neuron (N1) itself. Similar to our previous constraints, we induced phasic-like spiking through a current-step injection in our modeled neuron. The high spike rate induced an increase in intracellular calcium followed by a delayed onset GIRK conductance (Figure 4c2). Additionally, here, we assumed a quantal release concentration as in the previous scenario, i.e., an initial extracellular NE concentration of 200 nM. Despite the slowly rising GIRK conductance in N1, the calcium concentration in N1 only showed a subtle decline in the average intracellular calcium levels during these current-induced bouts of high neuronal activity (compare 4 c magenta traces upper versus lower trace). The spike rate in N1 remained stable despite a rising potassium leak through GIRK channels. Yet, we observed that after the current step subsided, the slow-decaying GIRK conductance impeded spontaneous firing in N1; 500 ms after the current pulse ended, spontaneous spiking recurred. Such an observation agrees with the typical quiescent period of LC neurons after episodes of high activity [44].

We then began to examine if somatic NE release and the spatial confinements of the LC architecture can give rise to volume-mediated interaction between LC neurons. Here, we hypothesized that the LC neurons during episodes of high activity can modulate spontaneous spiking of neighboring neurons via bystander inhibition. The spatiotemporal NE concentration profile originating from a point source can be estimated through Equation (13) [15,17].

The attenuation factor (*C*_0_/*C*(*r*,*t*)) relative to the initial NE concentration showed a spatiotemporal profile with an attenuation of 0.001 for distances of around 20 to 25 μm after 200 to 800 ms (Figure 5a). The fraction of NE bound alpha2-receptors (K_D_ = 1.25 nM) is shown as a temporal profile for various distances in Figure 5b. The onset and fractions of activated alpha2-receptor were slowed down and decreased through the time course of diffusion, respectively. In our neural model of LC neurons, we aimed to estimate the neighbor inhibition of freely diffusing NE onto a neuron (N3) at a distance of 25 μm. Evoking a phasic-like activity in neuron N1 through repeated current-step injections evoked an uprise in the intracellular calcium level that surpassed the threshold for NE vesicle release. The onset of a slow-rising GIRK conductance in N3 was delayed by 1400 ms compared to previous scenarios (dendro-somatic and autoinhibition) (compare Figure 4c1,c2 versus Figure 5c). The decelerated rise of GIRK conductance was due to the small number of active alpha2-receptors. This translated into a slow and small increase in GIRK conductance in N3. As GIRK activation was delayed, a second current step two seconds after the first one ended continued to build up GIRK conductance on the already existing conductance from the previous high activity bout (Figure 5c). Shorter intervals between the current step further facilitated this effect. This cumulated increase in GIRK conductance in N3 led to an enlarged interspike interval between spontaneous spikes in N3 (Figure 5c and Appendix A).

## 4. Discussion

Noradrenergic modulation exerts robust control over neuronal circuits throughout the entire brain. The densely packed core of the LC integrates various inputs from cortical, subcortical, and brain stem areas. These inputs are computed within the nucleus and evoke various network states in the LC. Early work highlights the importance of electronic coupling between noradrenergic neurons that can synchronize sub-threshold membrane fluctuation and spiking rates in the entire LC network [18,19,20,39], while later research points to multiple asynchronous neuronal ensembles that dynamically switch spiking patterns within the LC [3,14,45]. As the specific neuronal architecture and synaptic wiring onto and between LC neurons are unknown, the space of potential network states and their respective output function remains elusive. Recently, the identification of local inhibitory neurons that control the phasic activity of specific neuronal subsets in the LC adds another layer of computational capacity to this brain stem nucleus [46,47,48]. Yet, to achieve a deeper understanding of the functional role of noradrenergic subnetworks, we need experimental approaches that target these ensembles.

Here, we developed an experimentally validated neuronal model of noradrenergic cells that allowed us to investigate NE-mediated modulatory effects between LC neurons. First, our immunostaining against DbH and TH enzymes confirms the expression of both key enzymes for synthesize of NE in the vast majority of LC neurons in young and old mice (Figure 1). As recent studies demonstrated that noradrenergic axons in the dorsal hippocampus predominantly release dopamine [4,49], our results demonstrate that in somatic and dendritic compartments the NE can be produced in all LC neurons. Still, such a partition between dopamine and the NE release site could appear on the level of axonal varicosities, either through targeting of DbH enzymes to distinct projection areas or a differential expression of monoamine transporters that bias the uptake of a specific catecholamine from the extracellular space.

Our reconstruction of LC geometry reproduces noradrenergic cell numbers that are five times lower than earlier histological studies [50,51]. As previous studies use stereological cell counts of 6 to 50 μm brain slices, overestimation through double counting of neurons is possible. Yet, it is likely that these large differences are also due to insufficient penetration of TH antibodies in our cleared tissue preparation that limit the cell count compared to classical histology. Nevertheless, a possible underestimation of the cell count in our LC preparation would only affect the overall fraction of LC neurons that are in a 25 μm-radius, but the resolved absolute number of LC neuron pairs would stay constant or be larger.

In our diffusion model, we did not incorporate noradrenaline transporters activity (NET) that can limit the diffusion radius of NE. Despite studies showing the presence of transporters in the LC core [52,53], previous diffusion modeling studies for the Substantia nigra argue that the impact of monoamine transporter on the diffusion radius of dopamine is small [15,17]. Direct electrophysiological recordings in the LC neurons confirm that increasing electrical evoked-release of NE scales with the rate of alpha2-receptor evoked inhibitory potentials in noradrenergic neurons, arguing that monoamine transporters are not drawing excessive NE out of the extracellular space [54]. In contrast, in similar dose measurements in the ventral tegmental area, D1-receptor evoked inhibitory responses that did not show a dependence on stimulation strength. Only after treatment with a DAT antagonist were responses scaling with stimulation strength (Courtney and Ford 2014). These experimental findings argue that NET activity within the LC is not limiting the diffusion of NE.

The observed reduction in the spontaneous spike rate in our model through the diffusion of extracellular NE has so far not been described to our knowledge; it has been discussed as a plausible mechanism for dopaminergic neurons in the substantia nigra as well as dopaminergic axons [15,17,55]. Key factors for the observed bystander-inhibition are distances and geometrical orientation of the release site between LC neurons. Our model deduces that extracellular NE can accumulate through episodes of long-lasting phasic activity or repeated periods of bursting discharge. This accumulation is dependent on the speed of the G_i_ signaling cascade, the GIRK channel kinetic, and the dispersion of extracellular NE through the densely packed LC core. Such bystander inhibition adds another layer of interaction between LC neurons. The physiological role of such a type of inhibition is strongly interwoven with a topographical organization of the LC.

The idea of topographic organization of the LC has been put forward decades ago. Most studies at that time used local injection of retrograde dyes [56,57,58]. Despite controversial findings, it is generally accepted that the LC has a dorsal–ventral organization in which neurons in the dorsal part of the LC are preferential projecting to frontal areas, while the ventral part projects to brain stem areas. For an example, LC neurons that project to hippocampal areas or the olfactory bulb are more likely found in the dorsal part, while ventral LC neurons are more probable to project to the medulla or the spinal cord [57,59,60]. Based on new viral strategies and high-throughput axonal mapping technologies, a better understanding of the brain-wide input–output organization of the LC became available. These findings helped to develop new concepts that now allow us to understand the controversial findings from earlier tracing studies. For example, one of these concepts argues that axonal projections from a single LC neuron forms an extensive axonal field to a specific brain area while also possessing collaterals to other brain areas at lower density. A variation of such a concept entails the distinction between broadcasting LC neurons that form equally dense innervation to many areas versus exclusive LC neurons that form specific innervation to a single brain area. Both concepts would blur retrograde tracing experiments and underestimate a topographical organization of the LC. Interestingly, functional studies highlight heterogeneity of projection-defined LC neurons in terms of intrinsic excitability, after-hyperpolarization, and their proteomic make-up [13,29]. Here, LC neurons that preferentially project to the prefrontal cortex or hippocampus show a different sensitivity to alpha2-receptor agonists. Therefore, a topographic organization and differential expression of adrenergic receptors could fine-tune the impact of neighboring inhibition. It is interesting to speculate how the here-postulated volume-mediated interaction of spatially clustered LC neurons with different projection targets could amplify differences in extracellular NE in their respective axonal fields Figure 6. The increase in contrast of NE levels in two target regions could increase sensory saliency in one region.

## Figures and Tables

**Figure 1 brainsci-12-00728-f001:**
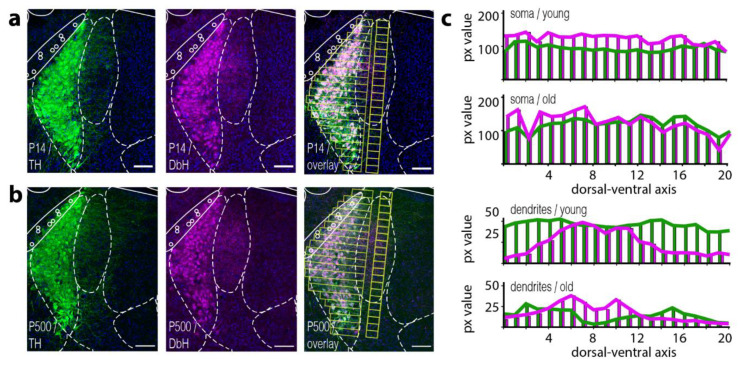
Tyrosine hydroxylase and dopamine-beta-hydroxylase expression largely overlap in somatic and dendritic compartments in the Locus coeruleus. (**a**) Shows immunostaining against key enzymes in the synthesis pathway of NE. White dotted lines show an overlay mask of the corresponding section from The Mouse Brain in Stereotaxic Coordinates 3rd Edition Franklin and Paxinos. (**a**) Confocal image of immunofluorescence against tyrosine hydroxylase (TH/green) from a brain stem slice of a young P14 mouse (anterior-posterior axis: −5.40 mm). The fluorescence signal is present in the soma as well as in the dendritic arbor (here more dominantly in the medial part); we observed a similar spatial distribution for the fluorescence signal from immunostainings against the dopamine-beta-hydroxylase (magenta). (**b**) Depicted are the same immunostainings as in (**a**) on brain slices obtained from an old P500 mouse. A similar spatial expression profile for both enzymes is visible when compared to slices from a young P14 mouse (**a**). (**c**) Shows a quantification of average pixel (px) intensities from the yellow-colored ROIs displayed in TH/DbH overlay image in (**a**,**b**). ROIs are placed above the somatic or dendritic compartments, respectively. (**c**) Shown is the average intensity of pixel values from ROIs along the dorsal–ventral axis for soma or dendrites. Scale bar 100μm.

**Figure 2 brainsci-12-00728-f002:**
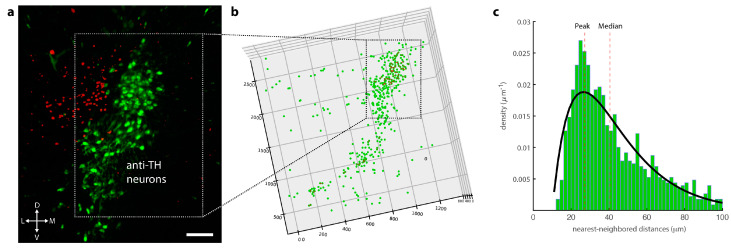
The largest fraction of noradrenergic neurons have neighboring LC neurons at a distance of 25 μm. (**a**) A typical fluorescent volume from a 2-mm cleared brain stem slice. Red dtomato fluorescence comes for parvalbumin-positive neurons while pixel intensities mapped in green at anti-TH/CY3 immunopositive neurons (scale bar 100 μm). (**b**) Cell segmentation was manually performed with IMARIS software and validated through two observers. The mean number of recovered LC neurons was 289 ± 45 (*n* = 4, 2 animals). Anti-TH positive neuron had a mean sphericity of 0.83 ± 0.12 and a mean volume of 4971 ± 0.233 μm^3^. (**c**) Nearest-neighbor distance histogram of pooled data sets from all four LC nuclei. Distances between neurons with less than 25 μm are connected through a red line. Nearest-neighbor distribution was fitted to a double exponential function (γ = 1.38, λ_long_ = 16.84 and λ_short_ = 16.23). The model predicts that the highest number of neurons are in bins between 24 and 28 μm. The peak of NN distribution and median are marked as dotted lines.

**Figure 3 brainsci-12-00728-f003:**
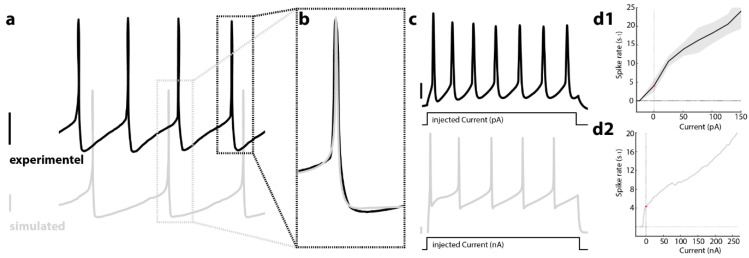
LC neuron model recapitulates essential spontaneous spiking, action potential characteristics, and current-evoked frequency dependency. (**a**) Spontaneous neuronal spiking in an LC neuron recorded in an acute brain slice from P70 mice via whole-cell patch-clamp (upper black trace). Action potentials show typical long after-hyperpolarization (AHP) that gave rise to a 4.2 Hz spontaneous spiking rate. Trace below shows simulated LC neuron that exhibit a similar spontaneous spiking rate and slow AHP. (**b**) Overlay of experimental and simulated action potentials exhibit a similar temporal profile. (**c**) The current-evoked increase in membrane potential evokes a barrage of action potentials in a recorded and simulated neuron (300 ms current injections). (**d1**) Illustrates the spike rate of recorded LC neurons under different current step amplitudes (error margin as SEM; *n* = 12 cells, 2 animals); (**d2**) Simulated LC neurons exhibit an increase in spike rate for different amounts of injected current. Note the different current units in (**d1**,**d2**) due to the different cell sizes of experimental and simulated neurons. Scale bars 10 mV for experimental neurons and 20 mV for simulated neurons.

**Figure 4 brainsci-12-00728-f004:**
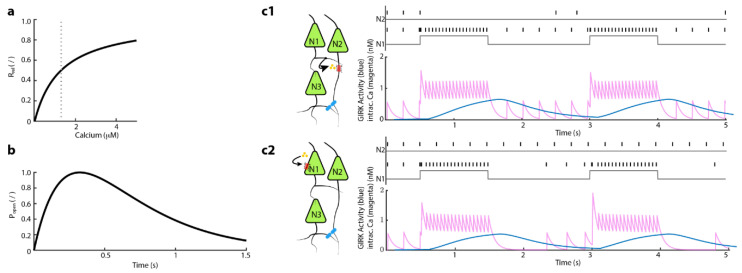
Simulation of spike rate from two LC neurons with a built-in NE-dependent GIRK activity leads to a reduced spiking rate in LC neurons that are connected via dendro-somatic synapses or a somatic release that retroact onto the secreting LC neuron. (**a**) *R_rel_* is the release fraction that is dependent on the level of presynaptic calcium at the time of action potential. The gray vertical line is the amount of calcium accumulated in a cell when firing at 20 Hz. The relation is defined by Equation (8). (**b**) *P_open_* shows the effective time course of the opening and closing of the synaptic conductance of GIRK channel (Equation (9) *P_open_*(*t*)). (**c1**) Depicts a simulation scenario where neuron 1 (N1) connects to neuron 2 (N2) via dendro-somatic synaptic interaction. The upper diagram shows the spike rates of N1 and N2 throughout the simulated 5-s. Here, the gray line represents the time course of current injections (1 s, I = 270 nA) in the respective neuron. The lower diagram shows the development of intracellular calcium levels in N1 (magenta) during a phasic-like activation; the blue line represents the time course of GIRK conductance in N2. (**c2**) Cartoon illustrates the autoinhibition scenario in which alpha2-adrenergic autoreceptors in N1 activate through local NE-release from the soma. One-second bouts of current injections (I = 270 nA) led to an increased spike rate in N1, while N2 displays a spontaneous firing pattern (upper diagram). The lower diagram shows time courses of intracellular calcium concentration and GIRK conductance in N1.

**Figure 5 brainsci-12-00728-f005:**
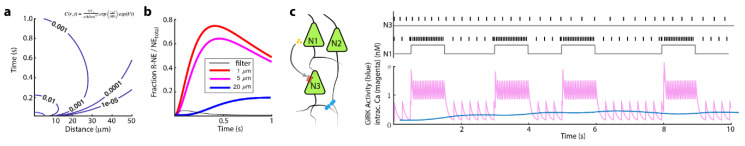
Spike modulation in the neighboring neuron of somatically release NE cumulatively affects neighboring neurons within a 25 μm radius. (**a**) Spatiotemporal diffusion profiles of NE from a point source. The diffusion constant for NE was 3.4 × 10^−6^ cm^2^ s^−1^ [15,17]. (**b**) Time course of fractions of NE bound alpha2-receptor at 1, 5, and 20 μm distances from release site. Kinetics of transmitter binding and release was modeled by an alpha function (think black line, *τ* = 50 ms). (**c**) Simulation of spike train of current injected neurons (N1) and nearby neurons (N3/25 μm distance). The lower plot shows the development of intracellular calcium levels in N1, while the blue line represents GIRK conductance in N3.

**Figure 6 brainsci-12-00728-f006:**
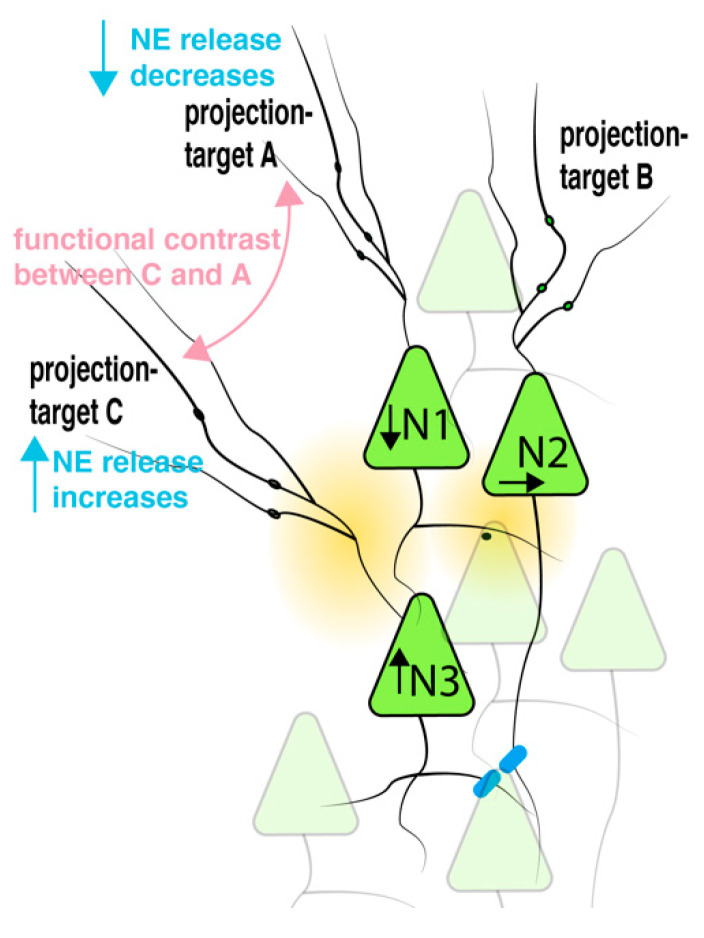
Illustration of functional consequences of a volume-mediated inhibition of neighboring neurons with different projection targets. Strong enduring input to neuron 3, N3, led to an increase in neuronal spiking (arrow up) and a subsequent increased release of NE within the core of the LC. Based on distances and receptor portfolio of neuron 1 (N1) from the release sites of N3, the spontaneous spiking activity in N1 was decreased through activation of GIRK (arrow down), while, for example, activity in neuron 2 (N2) was not altered (arrow horizontal). Consequently, axonal terminals from N3 to projection-target area C increased spike-dependent release of NE, while axons in area A from N1 decreased NE release due to decreased spiking in N1. It is conceivable that the volume-mediated local NE inhibition in the core of the LC led to contrasting functional effects in the neuronal circuits in region C and A, i.e., facilitating sensory saliency in region C versus region A.

## Data Availability

All code used for all analyses and plots are publicly available on 484 GitHub at: https://github.com/teamprigge, Last accessed on 12 May 2022.

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
