# Peer review of "Spike-Dependent Dynamic Partitioning of the Locus Coeruleus Network through Noradrenergic Volume Release in a Simulation of the Nucleus Core"

_brainsci, 2022, doi:10.3390/brainsci12060728_

Round 1
Reviewer 1 Report
In this paper, Barai et al. examined physiological activities of noradrenergic neurons in the mouse locus coeruleus and suggested that several wiring patterns within the core locus coeruleus. To date, many papers have shown that biological significances of noradrenergic circuits from the locus coeruleus toward other brain regions, such as the hippocampus and cerebral cortex. On the other hand, information regarding neural circuits within the locus coeruleus is still limited. Thus, this reviewer is thinking that the authors' studies are valuable in the field of neuroscience. However, this manuscript has some problems that should be solved.
Major comments;
- Need to describe the mouse strain the authors used in M&M (page 3, line 12).
- As the authors mention in the discussion, the difference in neuronal features, such as axonal projection, between dorsal and ventral regions in the LC is an important issue. Thus, it is necessary to show where in LC subregions the authors examined. Please add describing a subregion(s) you focused on in M&M (dorsal or ventral? lateral or medial?).
- In the Results, this reviewer does not understand why the authors used mice in young, adult, and old ages (i.e., P14 and P500 in Fig. 1; older than P70 in Fig. 3). The authors have to show rational explanations about why you chose such different ages. In addition, as you used multiple ages, their ages in each figure are confusing. Why don't you describe their ages in the figure legends?
- Some researchers have suggested sexual dimorphic of the LC. Thus, the authors should describe sexes of the mice used in experiments, which will be helpful for other researchers.
- This reviewer is wondering about what is something new in the Fig. 1. TH and DbH are the enzymes essential for NE synthesis and well-known marker proteins in LC-NE neurons. The other problems in the Fig. 1. are that the authors did not mention "a3" and "c1, c2" in the legend and what the dotted-white lines in the a1-3 and b1-3 are. Additionally, I am wondering about why the illustration in Fig. 1e is shown here. Is there any rational relationship between Fig. 1a-c and Fig. 1e? Furthermore, what is the light-blue stuffs on right bottom in the drawing in e? Collectively, I recommend you to carefully revise the Fig. 1.
- In the Fig. 4b, although there are four-colored lines in the graph, its legend shows only three. In addition, the color of the line indicating the "1 um" condition is not clear (that is too thin).
- The authors mentioned LC-associated brain functions and diseases in the introduction; however, your discussion does not talk about a relationship of authors' results with LC functions. The authors need to add some discussion about how your results contribute to better understanding of brain functions and diseases.
Minor comments;
- Please spell out "NE" in the abstract.
- Is the degree of the y-axis in Fig. 2c correct? Because the authors examined the cells based on the x-y-z coordinates, "/um3" might be properly...
- "see figure 2c" should be correct in page 9, line 5.
Author Response
- Need to describe the mouse strain the authors used in M&M (page 3, line 12).
Thank you for the reviewers comments .We added the corresponding information in M&M.
- As the authors mention in the discussion, the difference in neuronal features, such as axonal projection, between dorsal and ventral regions in the LC is an important issue. Thus, it is necessary to show where in LC subregions the authors examined. Please add describing a subregion(s) you focused on in M&M (dorsal or ventral? lateral or medial?).
It is an important point the reviewer raises here. For the analysis of our DbH- and TH immunostaining, we only analysed a the dorsal-ventral axis. We also find a similar pattern of DbH/TH distribution along the anterior-posterior axis. We added representative figure as Supplement 3 to the manuscript.
We added the exact coordinates that we used for the overlay in figure 1 form The Mouse Brain in Stereotaxic Coordinates 3rd Edition Franklin & Paxinos into the figure legend as well as M&M.
- In the Results, this reviewer does not understand why the authors used mice in young, adult, and old ages (i.e., P14 and P500 in Fig. 1; older than P70 in Fig. 3). The authors have to show rational explanations about why you chose such different ages. In addition, as you used multiple ages, their ages in each figure are confusing. Why don't you describe their ages in the figure legends?
The reason why we used young and old animals in figure 1, is to address the view that the protein make-up is drastically changed in noradrenerigc neurons during the postnatal phase compared with adult or old animals e.g. changing of connexin subunits or NaV subunits.
In figure 1, we argue that the enzymes for NE production are not affected by the reorganisation of the protein portfolio in noradrenergic neurons during development. Meaning that all noradrenergic neurons are likely to produce NE at any stage during their lifetime, and are therefore capable of inducing neighboring inhibition.
For our electrophysiological measurements, we used P70 (10-week) for all our experiments within the lab for reason of comparability.
We followed the reviewers suggestion and added the age into the figure legends.
- Some researchers have suggested sexual dimorphic of the LC. Thus, the authors should describe sexes of the mice used in experiments, which will be helpful for other researchers.
Indeed, a very interesting and relevant aspect. We are looking into the distribution of TH and DbH in female mice right now. As an undergraduate student project. We are planning to make these information public to the community as soon as we have our results.
- This reviewer is wondering about what is something new in the Fig. 1. TH and DbH are the enzymes essential for NE synthesis and well-known marker proteins in LC-NE neurons. The other problems in the Fig. 1. are that the authors did not mention "a3" and "c1, c2" in the legend and what the dotted-white lines in the a1-3 and b1-3 are. Additionally, I am wondering about why the illustration in Fig. 1e is shown here. Is there any rational relationship between Fig. 1a-c and Fig. 1e? Furthermore, what is the light-blue stuffs on right bottom in the drawing in e? Collectively, I recommend you to carefully revise the Fig. 1.
These concerns partially overlap with question 3. Additionally, the new and surprising finding is the spatial distribution of the DbH and TH enzymes within the dendritic compartment, and also the lower expression of DbH on the anterior and posterior pols is not described in literature to our knowledge. In the argumentation for our hypothesis of an NE-mediated inhibition of neigbouring noradrenergic neurons, it is important to confirm the presence of both relevant enzymes in the soma throughout the entire LC. Our Immunostaing suggests that not all LC neurons produce the same amount of NE.
We explained dotted-white lines in figure a1-3 and b1-3
As we agree with the reviewer for the need of improvement for the numbering in figure 1, we revised the entire image. As figure 1 e was intended to introduce the concept and the different scenarios that we explored throughout the manuscript, we now believe that an improved version of figure 1 e could serves a summary figure. We included the improvet figure 1e as separate summary figure 6.
- In the Fig. 5b, although there are four-colored lines in the graph, its legend shows only three. In addition, the color of the line indicating the "1 um" condition is not clear (that is too thin).
Thank you for pointing that out! We corrected the legend. Coloured lines show the fraction of bound receptors relative to total NE at different distances. The thin black line shows the temporal profile of our filter that accounts binding and release of NE.
- The authors mentioned LC-associated brain functions and diseases in the introduction; however, your discussion does not talk about a relationship of authors' results with LC functions. The authors need to add some discussion about how your results contribute to better understanding of brain functions and diseases.
Indeed this is a very relevant and justified question that is basically to ask for the functional relevance of our hypothesis. And indeed, during the process of writing the manuscript, we decided not to address the functional implication in our discussion due to the lack of direct experimental verification of such an effet. Yet, due preprint publication we got in contact to servaral researchers that studied similar effect with different epxerimental designs. That being said, we are now more confident to discuss consequences of our theoretic model. We include a revised schematic figure (figure 6) that illustrates the functional implication of our proposed theory.
Minor comments;
- Please spell out "NE" in the abstract.
corrected
- Is the degree of the y-axis in Fig. 2c correct? Because the authors examined the cells based on the x-y-z coordinates, "/um3" might be properly…
It is the euclidan distance between two point in 3D, therefore mum is correct
- "see figure 2c" should be correct in page 9, line 5.
corrected
Reviewer 2 Report
Recent research revealed an unexpected cellular input specificity within the nucleus that can give rise to various network states that either broadcast norepinephrine signals throughout the brain or pointedly modulate specific brain areas. The distances between noradrenergic neurons in the core of the Locus coeruleus are unusually small, neighboring neurons could theoretically impact each other via volume transmission of norepinephrine. Thus, the aim of this study was to investigate bystander-inhibition via noradrenergic alpha2-receptors through a computational neuron model. The authors validated the neuron model through comparison with experimental patch-clamp data and identified key variables that impact bystander-inhibition of neighboring neurons. Their spike model of neurons predicted that repeated or long-lasting episodes of high neuronal activity induce a bystander-inhibition that reduces the spike rate in neighboring neurons at distances smaller than 25 micrometers. The authors conclude that their findings can guide future experimental approaches to test this phenomenon and its physiological consequences. The manuscript is well-written and the methods sound. I have some comments that I believe need to be addressed.
Comments:
The authors used mice for acute brain slice preparation and immunohistochemistry. However, I failed to find any information that the animal experiments were approved by the local ethical commission. Approval. no. should be also included in the manuscript.
Page 1 lines 13–15, “Recent research revealed an unexpected cellular input specificity within the nucleus that can give rise to various network states that either broadcast NE signals throughout the brain or pointedly modulate specific brain areas.”, Please define “NE”.
Page 3 lines 4–6, “These high calcium levels can trigger the release of NE-containing vesicles from dendritic and somatic compartments (Hong-Ping Huang et al. 2012; H.-P. Huang et al. 2007).”, Huang et al. 2007 is not in references list.
Page 4 line 23, “Afterward, samples were washed in in PBS…”, Please revise this sentence.
Page 10 lines 6–8, “High spiking rates lead to membrane fusion of NE-containing large-dense vesicles at the soma (H.-P. Huang et al. 2007; Hong-Ping Huang et al. 2012).”, Huang et al. 2007 is not in references list.
Page 13 lines 6–8, “As recent studies demonstrated that noradrenergic axons in the dorsal hippocampus predominantly release dopamine (Kempadoo et al. 2016; Takeuchi et al. 2016),”, Kempadoo et al. 2016 is not in references list.
Page 13 lines 14–16, “Our reconstruction of LC geometry reproduces noradrenergic cell numbers that are five times lower than earlier histological studies (Sturrock and Rao 1985; O’Neil et al. 2007).”, Sturrock and Rao 1985 and O’Neil et al. 2007 are not in references list.
Page 14 line 1, “more probabil”
Page 14 lines 15–16, “Here, LC neurons that preferentially project to PFC or hippocampus show a different sensitivity to alpha2-receptor agonists.”, Please define “PFC”.
References 36 is not quoted.
Author Response
The authors used mice for acute brain slice preparation and immunohistochemistry. However, I failed to find any information that the animal experiments were approved by the local ethical commission. Approval. no. should be also included in the manuscript.
Thank you for the reviewers comments .We added the corresponding information in M&M.
Page 1 lines 13–15, “Recent research revealed an unexpected cellular input specificity within the nucleus that can give rise to various network states that either broadcast NE signals throughout the brain or pointedly modulate specific brain areas.”, Please define “NE”.
Corrected
Page 3 lines 4–6, “These high calcium levels can trigger the release of NE-containing vesicles from dendritic and somatic compartments (Hong-Ping Huang et al. 2012; H.-P. Huang et al. 2007).”, Huang et al. 2007 is not in references list.
Corrected
Page 4 line 23, “Afterward, samples were washed in in PBS…”, Please revise this sentence.
Corrected
Page 10 lines 6–8, “High spiking rates lead to membrane fusion of NE-containing large-dense vesicles at the soma (H.-P. Huang et al. 2007; Hong-Ping Huang et al. 2012).”, Huang et al. 2007 is not in references list.
Corrected
Page 13 lines 6–8, “As recent studies demonstrated that noradrenergic axons in the dorsal hippocampus predominantly release dopamine (Kempadoo et al. 2016; Takeuchi et al. 2016),”, Kempadoo et al. 2016 is not in references list.
Corrected
Page 13 lines 14–16, “Our reconstruction of LC geometry reproduces noradrenergic cell numbers that are five times lower than earlier histological studies (Sturrock and Rao 1985; O’Neil et al. 2007).”, Sturrock and Rao 1985 and O’Neil et al. 2007 are not in references list.
Corrected
Page 14 line 1, “more probabil”
Corrected
Page 14 lines 15–16, “Here, LC neurons that preferentially project to PFC or hippocampus show a different sensitivity to alpha2-receptor agonists.”, Please define “PFC”.
Corrected
References 36 is not quoted.
Now - all references are quoted
Reviewer 3 Report
Paper titled (Exploring volume mediated bystander-inhibition in a neuron model of the Locus coeruleus) by Barai et al. provides a method for exploring the bystander inhibition in Locus coeruleus model. Although the authors did a good effort for validating the method, I noticed some corrections are mandatory.
1- Title: needs revision as it is not clear or informative & the reader cannot guess the influence of the inhibition in the neuronal cells.
2- Immunohistochemistry : mention the code number of antibodies and kits
3- Immunohistochemisyrt: mention the dilution of the antibody
4- Give the origin of the chemicals completely and consistently: code number, company, town, state and country, allover the manuscript;
5- Do the same for software (inclue also the version)
6-Introduction is long and needs to be shortened to be more concrete. One part consists of a very long paragraph and should be spletted into relevant parts.
7- Did the authors perform any repetetion for this methodology to confirm the findings? Any statistical analysis was done to validate the method?
Author Response
1- Title: needs revision as it is not clear or informative & the reader cannot guess the influence of the inhibition in the neuronal cells.
Thank you for that opinion. Similar concerns were raised for our peers. We change the title accordingly.
2- Immunohistochemistry : mention the code number of antibodies and kits
Corrected
3- Immunohistochemisyrt: mention the dilution of the antibody
Corrected
4- Give the origin of the chemicals completely and consistently: code number, company, town, state and country, allover the manuscript;
Corrected
5- Do the same for software (inclue also the version)
Corrected
6-Introduction is long and needs to be shortened to be more concrete. One part consists of a very long paragraph and should be spletted into relevant parts.
We revised and shortened the introduction.
7- Did the authors perform any repetetion for this methodology to confirm the findings? Any statistical analysis was done to validate the method?
All experiments have been repeated at least twice with at least two independent experiments/animals. Where possible parametric statistics (assuming standard distribution) has been applied
Round 2
Reviewer 3 Report
the revised version of paper titled (Spike-dependent dynamic partitioning of the Locus coeruleus network through noradrenergic volume release in a simulation of nucleus core). was revised adequately. I recommned publication of the current form.